# Urinary Angiotensinogen Displays Sexual Dimorphism in Non-Diabetic Humans and Mice with Overweight

**DOI:** 10.3390/ijms25010635

**Published:** 2024-01-04

**Authors:** Alexis A. Gonzalez, Bruna Visniauskas, Virginia Reverte, Ventaka N. Sure, Zoe Vallotton, Bryan S. Torres, Marco A. Acosta, Mahlet Zemedkun, Prasad V. Katakam, Minolfa C. Prieto

**Affiliations:** 1Instituto de Química, Pontificia Universidad Católica de Valparaíso, Valparaíso 2340025, Chile; 2Department of Physiology and Hypertension Core, Tulane University School of Medicine, New Orleans, LA 70112, USA; 3Department of Pharmacology, Tulane University School of Medicine, New Orleans, LA 70112, USA; 4Renal and Hypertension Center of Excellence, Tulane University School of Medicine, New Orleans, LA 70112, USA

**Keywords:** kidney injury markers, high-fat diet, sex differences, telemetry

## Abstract

Increased body weight (BW) induces inappropriate renin–angiotensin system (RAS) activation. The activation of the intrarenal RAS is associated with increased urinary angiotensinogen (uAGT), blood pressure (BP), and kidney damage. Here, we examined uAGT excretion levels in young non-diabetic human subjects with overweight (OW) and non-diabetic mice with high-fat diet (HFD)-induced OW. Human subjects (women and men; 20–28 years old) included two groups: (a) overweight (OW, *n* = 17, BMI ≥ 25); and (b) controls (normal weight (NW; *n* = 26, BMI ≤ 25). In these subjects, we measured BP, albuminuria, and protein levels of uAGT by ELISA adjusted by urinary creatinine (expressed by uAGT/uCrea). Mice (female and male C57BL/6J mice, 8 ± 2 weeks of age) also included two groups: HFD or normal fat diet (NFD) fed for 8 weeks. We measured BW, fasting blood glucose (FBG), BP by telemetry, albuminuria, and uAGT by ELISA. In humans: (i) no significant changes were observed in BP, albuminuria, and FBG when comparing NW and OW subjects; (ii) multivariate logistic regression analysis of independent predictors related to uAGT/uCrea levels demonstrated a strong association between uAGT and overweight; (iii) urinary reactive oxygen species (ROS) were augmented in men and women with OW; (iv) the uAGT/uCrea ratio was higher in men with OW. However, the uAGT/uCrea values were lower in women even with OW. In mice: (i) males fed an HFD for 8 weeks became OW while females did not; (ii) no changes were observed either in FBG, BP, or albuminuria; (iii) kidney ROS were augmented in OW male mice after 28 weeks but not in females; (iv) OW male mice showed augmented excretion of uAGT but this was undetectable in females fed either NFD or HFD. In humans and mice who are OW, the urinary excretion of AGT differs between males and females and overcomes overt albuminuria.

## 1. Introduction

It is well established that being overweight (OW) predisposes to cardiovascular (CV) and renal diseases, including diabetes mellitus (DM) [1]. During diabetes, the intrarenal renin–angiotensin system (RAS) is inappropriately activated as reflected in the stimulation of intrarenal renin and prorenin synthesis, angiotensin-converting enzyme (ACE), and intratubular levels of angiotensinogen (AGT) [2,3]. The intrarenal and intratubular RAS contribute to the development of hypertension through increasing intratubular angiotensin II (AngII)-dependent activation of Na^+^ transporters [4]. Patients with DM develop hypertension and CV complications and an increased risk of chronic kidney disease (CKD) [5]. Blockade of the RAS mitigates the effects of abnormal intrarenal RAS activation in humans and rodents with hypertension and or diabetes [6,7]. AGT is the substrate for renin and is primarily formed by the liver; however, it is also synthesized and secreted by the proximal tubule cells in the kidneys [8]. During diabetes and hypertension, AGT transcription and protein synthesis are augmented [9,10]. Indeed, urinary AGT levels (uAGT) can be used as an indicator of RAS activation in diabetes [11]. Because uAGT levels precede the onset of albuminuria—a primary indicator of RAS activation—in normotensive patients with type 2 DM (T2DM), it has been suggested that uAGT may potentially serve as an early marker of intrarenal RAS activation and progressive CKD in T2DM patients without hypertension. Nevertheless, the relationship between uAGT and albuminuria and the impact on kidney injury in OW and prediabetic conditions is unknown. Urinary mRNA levels of AGT and kidney injury markers such as interleukin-18 (IL-18), connective tissue growth factor (CTGF), neutrophil gelatinase-associated lipocalin (NGAL), and kidney injury molecule-1 (KIM-1) correlated with increased body mass index (BMI) in young human adults. In these subjects, systolic blood pressure (SBP), fasting blood glucose levels (FBG), and urinary protein also correlated with BMI [12], suggesting that injury and inflammatory kidney markers are augmented in obesity and are associated with increased oxidative stress [13,14].

Obesity linked to pre-diabetic conditions may affect redox status in kidney cells and favor cell signaling pathways that enhance the extracellular matrix and damage renal tubules [15,16]. Male mice fed a high-fat diet (HFD) for 28 weeks developed type 2 diabetes with augmented uAGT excretion and kidney ROS [17]. The abnormal ROS augmentation in renal tubular epithelial cells may be a mechanism explaining renal tubular injury [18,19]. Therefore, further studies using mouse models of obesity and diabetes and clinical observations from humans are required as a prerequisite to clinical trials on new targets to treat CKD in diabetes. In the present study, we examined whether urinary protein AGT levels are increased in non-diabetic humans and mice with OW. We further analyzed the correlations of uAGT with BP, BMI, and albuminuria in subjects of both sexes for further insights into the impact of sex as a biological variable on the progression of metabolic disease in mice and humans with OW.

## 2. Results

### 2.1. Results in Humans

#### 2.1.1. Clinical Characteristics and Urinary AGT Levels in Normal and Overweight Subjects

Figure 1 shows some of the clinical data in the population. Among the subjects participating in the study, 17 individuals were OW (9 men and 8 women), and 26 individuals had NW (13 men and 13 women). The inclusion and exclusion criteria are presented in Table 1. SBP showed a slight, but not significant, increase compared to NW men (118 ± 3 vs. 112 ± 3 mmHg) and women (121 ± 3 vs. 112 ± 2 mm Hg). Similarly, DBP was slightly elevated in OW women (78 ± 8 vs. 68 ± 6 mmHg). FBG in OW men was not different from NW (95.1 ± 2 versus 90.7 ± 1 mg/dL). The same observation was seen in women (OW: 91.8 ± 3 versus 90.1 ± 2 mg/dL). Albuminuria measurements yielded no observable changes in OW vs. NW subjects or between sexes. Inclusion criteria and exclusion criteria are reported in Table 1.

#### 2.1.2. Urinary AGT Levels in Normal and Overweight Subjects and Possible Predictors of High Levels of uAGT

Urinary AGT vs. creatine ratio was segmented in men with OW. The uAGT/uCrea ratio was significantly lower in women with NW and OW when compared to men (Figure 1). We performed simple linear regression analysis for uAGT/uCrea vs. SBP showing no significant correlation in men (deviation from zero: 95% confidence intervals: slope: −0.02333 to 0.1198, R^2^: 0.0564). The correlation analysis in women showed similar results (deviation from zero: 95% confidence intervals: slope: 0.011232 to 0.11211, R^2^: 0.0322). When uAGT/uCrea was analyzed with BMI, we found a significant correlation for AGT vs. BMI in men (deviation from zero: 95% confidence intervals: slope: 0.05455 to 0.32122, R^2^: 0.5666, slope significant from zero? = *p* < 0.01) and women (deviation from zero: 95% confidence intervals: slope: 0.02322 to 0.4530, R^2^: 0.567, slope significant from zero? = *p* < 0.01).

We next performed a multivariate logistic regression analysis to identify the independent predictors of uAGT/uCrea ratio. Logistic regression was used to evaluate the response (outcome) variable to predict the probability of this outcome occurring. We included gender (male or not) as a predictor; fasting blood glucose (FBG) less than 100 mg/dL; systolic blood pressure (SBP) less than 120 mm Hg, diastolic blood pressure (DBP) less than 80 mm Hg; and overweight (BMI ≥ 25), see Table 2.

#### 2.1.3. Reactive Oxygen Species in Urine

We next evaluated ROS levels in urine samples from normal and overweight subjects. As shown in Figure 1, levels of urinary 8-Hydroxydeoxyguanosine (8-OHdG) were augmented in OW men and women. Since no established high or normal levels of ROS have been described in the literature, we did not categorize 8-OHdG values as high or low for logistic regression analysis.

### 2.2. Results in Mice

#### Physiological Characteristics and Urinary Angiotensinogen in Females and Males in Mice with Control and High Fat Diet

We next evaluated body weight, FBG, urinary AGT (by ELISA), albuminuria, and blood pressure in female and male mice subjected to 8 weeks of HFD or CD (Figure 2). The previous application of this protocol yielded OW mice after 4–8 weeks of HFD and development of Type 2 diabetes at 16 weeks [17]. HFD mice became OW with significant increases in BW in males (31 ± 2 vs. 24 ± 1 g; *p* < 0.0001) but not in female mice (20 ± 1 vs. 19 ± 2). BW was different between males and females fed CD (24 ± 1 vs. 19 ± 2, *p* < 0.05, 13). Mice of the same sex but different diets showed no change in FBG; however, a significant difference was found between males and females fed CD. uAGT was augmented in males fed HFD as compared to controls (4.47 ± 0.7 vs. 0.43 ± 0.2 ng/day, *p* < 0.05); however, uAGT was undetectable in female mice fed CD or HFD. Albuminuria was not augmented with diet when analyzed in males or females and no statistical differences were found between males and females fed CD or HFD. No significant differences were found in SBP and DBP.

## 3. Discussion

In the present study, we demonstrate that in humans with OW the uAGT levels are augmented in both women and men; however, the increase was less evident in women. Increases in uAGT were also in parallel with increased levels of urinary ROS. In mice with HFD diet-induced OW, the uAGT were elevated in males but undetectable in female mice. In both human subjects and mice, the increases in uAGT were not in parallel with albuminuria.

In our mouse model of OW without diabetes, female mice showed undetectable levels of uAGT in urine. Previous studies in rats have shown that a high salt diet greatly exacerbated the uAGT excretion in Ang II-infused Sprague Dawley rats with higher increases in males (9-fold increase over Ang II alone) than in females (2.5-fold increase). Also, male rats displayed salt-sensitive SBP increases during Ang II infusion and an HS diet, and female rats did not [20]. In human subjects, the uAGT/uCreat ratio was reduced in women versus men with NW. This was consistent with the results observed in mice. We have recently reported that the augmentation of uAGT was associated with high levels of soluble (pro)renin receptor (sPRR) in men but not in women with type 2 diabetes mellitus [17]. The PRR is involved in the progression of diabetic kidney disease highlighting the relevance of sPRR as a potential biomarker of kidney disease [21,22,23,24]. Indeed, uAGT is significantly higher in those patients with higher levels of plasma sPRR than in patients with low levels of uAGT. This difference was significant only in men [20]. Similarly, an HFD for 28 weeks in male mice leads to a T2DM phenotype and concomitant increased sPRR in plasma, suggesting that sPRR may indicate the status of systemic RAS activation and the onset of vascular complications during T2DM in a sex-dependent manner [24]. These findings are important, as Sartori-Valinotti et al. demonstrated that females are protected from renal injury when given Ang II and high salt, whereas the increase in renal injury and oxidative stress in males may play a role in exacerbation of hypertension with Ang II and high salt. Furthermore, AGT expression increases in the renal cortex of male rats treated with Ang II and high salt, but not in females [25].

Obesity is a significant risk factor for diabetes, hypertension, and renal diseases. Obesity is associated with proteinuria and loss of nephron function which exacerbates hypertension [26]. In the kidneys, the activation of the RAS has also been associated with the augmentation of uAGT, increased SBP, and the development of CKD [27,28]. In the present study, despite the significant increase in BMI, we did not detect changes in SBP or albuminuria in OW and control lean human subjects, which is consistent with a non-diabetic phenotype. In the human subjects of the present study, uAGT correlated with BMI, although it seemed to be less prominent in women, and the multivariate logistic regression analysis of independent predictors showed no significant association with sex when the analysis was categorized by gender. This seems to be more evident in mice according to sex; however, the total number of mice did not allow us to perform this type of analysis. In rats, it has been shown that there are significant sex differences regarding renal and urinary AGT levels showing lower expression of mRNA and protein levels in the kidneys of female rats and lower levels of uAGT [29]. Accumulated evidence has demonstrated that AGT is increased via the administration of estrogen, whereas renin, angiotensin-converting enzyme (ACE), and AT-1 receptor are down-regulated. Under some experimental conditions, estrogen appears to suppress the RAS [30].

Mice fed HFD showed an increased SBP starting at 20 weeks on this diet [17]. In the present study, we demonstrated that male mice with OW exhibited increased urinary AGT before the development of hypertension and kidney injury. Nonetheless, in female mice—either OW or lean—levels of uAGT were undetectable. Of note, we demonstrated that there were undetectable levels of uAGT in female mice fed HFD and in female mice. This may be related to previous data in rats that demonstrated male rats display salt-sensitive hypertension during Ang II infusion and HS diet; however, female Sprague Dawley rats do not show this response [31]. In our study, women with OW showed a modest increase in SBP and DBP.

The overactivation of intrarenal and intratubular RAS favors the development of hypertension and kidney damage during diabetic nephropathy [32,33]. Even when diabetic patients have normal to low levels of plasma renin activity, RAS inhibitors are still effective in reducing albuminuria; thus, supporting the role of the RAS in these compartments in the progression of diabetic nephropathy. The evidence indicates that increased proximal tubular secretion AGT in humans and rodents with hypertension and diabetes [9,28,34,35] contribute to the intratubular augmentation of intratubular Ang II formation, as AGT spilled over the distal nephron serves as a substrate for prorenin filtered or produced by the collecting duct, which increases sodium reabsorption and blood pressure [2]. Indeed, even when PRA is normal or suppressed in diabetic patients, levels of prorenin in plasma are augmented [36,37]. The presence of ACE and the (pro)renin receptor in the collecting duct further supports the contribution of the RAS toward the progression of high blood pressure and tissue fibrosis in diabetic kidneys.

The activation of the intratubular RAS promotes the synthesis of profibrotic markers through the Ang II type 2 receptor [38] and PRR by locally synthesized and filtered prorenin [39]. Gilbert et al. quantified the urinary excretion of the profibrotic cytokine connective tissue growth factor (CTGF) in patients with type I diabetes and incipient and overt diabetic nephropathy. Also, CTGF excretion was related to the severity of diabetic nephropathy, suggesting that urinary CTGF may provide a marker of renal injury [40]. In a cross-sectional study, we reported that the mRNA levels of urinary kidney injury markers such as interleukin-18 (IL-18) and CTGF were augmented in individuals with a high BMI [12]. However, we did not find any changes in the neutrophil gelatinase-associated lipocalin (NGAL) and kidney injury molecule-1 (KIM-1) mRNA levels [12]. Terami et al. showed that uAGT positively correlated with the albumin/creatinine ratio and urinary α1-microglobuline and pointed out that uAGT–α1-microglobulin correlation more accurately reflected the tubular injury associated with the intrarenal RAS activation in patients with T2DM [41].

There is an association between uAGT and cardiovascular and renal diseases. The Bogalusa Heart Study showed that in asymptomatic young adults encompassing a biracial subject population, the uAGT/uCreat ratio is positively correlated with SBP and DBP, but not with BMI or serum creatinine levels, independent of race or sex. The uAGT correlation with SBP and DBP was particularly strong in black male subjects [28]. Sawaguchi et al. described the correlations between uAGT and several clinical parameters, such as albuminuria and glomerular filtration rate, but not between uAGT and plasma angiotensinogen. These associations support the concept that uAGT comes primarily from proximal tubule origin [42] and that the increases in uAGT reflect intrarenal activation of the RAS.

Limitations of the study. The present study in humans was designed as a non-invasive protocol allowing the obtention of fresh urine samples and the protocol allowed the participation of a cohort of young men and women with no associated pathologies. The protocol did not include blood analyses for the determination of creatinine levels, estimated glomerular filtration rate, and other components of the systemic RAS. Further studies are needed to examine the associations between uAGT protein levels and injury markers TGF-b and a-SMA in humans and mice. The association between BMI and uAGT in the absence of albuminuria in human subjects requires further investigation, taking advantage of new methods of uAGT detection in urine. Further studies to examine the underlying hormonal mechanisms involved in the reduced levels of uAGT in female mice and women are warranted.

In conclusion, the levels of uAGT display sexual dimorphism and are associated with BMI in men and women with overweight. In this cohort, no significant changes in BP, albuminuria, and FBG were found when comparing NW and OW human subjects. Consistently, male mice with HFD-induced OW show augmented uAGT levels, while in female mice, uAGT is undetectable. In mice with overweight, urinary levels of ROS are augmented in both males and females, but they do not exhibit changes in FBG, BP, and albuminuria. Therefore, increases in uAGT may reflect a differential subclinical activation of the intrarenal RAS in individuals with overweight in a sex-different fashion.

## 4. Materials and Methods

### 4.1. Studies in Humans

Study population: A cohort of 51 subjects was recruited. This study was approved by the Ethics Committee of the Pontifical University of Valparaiso, Chile (CODE BIOEPUCV-H-205-2018). Each participant was given an informed consent form; this provided adequate information to allow them to make an informed decision about participation in the clinical study, which included the collection of urine samples, and the confidentiality of the analysis and results. Of the 51 subjects, eight patients were subsequently excluded due to associated diseases. Inclusion criteria are described in Table 1.

Anthropometric measurements: Measurements of body weight, BMI, and height were performed on the day of biochemical analysis. According to the BMI (kg/m^2^) measurements, we divided the total group into two: NW (BMI ≤ 25.0) or OW (BMI ≥ 25.0).

Blood pressure measurements: BP measurements were performed according to the auscultatory method using a certified mercury sphygmomanometer in a quiet room with a comfortable temperature. The subject was asked not to talk during measurements. Each subject’s arm was bare and placed at heart level. A standard cuff (12.5 cm bladder) was used and fitted appropriately to the subject’s arm size. Systolic and diastolic blood pressure was recorded as the mean value of 3 measurements.

Urine analysis: Urine samples (20–30 mL) were collected in the morning. The urine was deposited in a sterile container and kept at 4 °C in the laboratory for subsequent analysis. Samples were discarded if they contained bacteria. Urinary albumin concentration was measured after a 10-min sample centrifugation using a human albumin ELISA Kit (ab108788) and converted to mg per day. Urinary 8-Hydroxydeoxyguanosine (8-OHdG) was measured with ELISA as a quantitative detection of the oxidized DNA adduct 8-OHdG in urine. Measurements were performed according to the manufacturer’s instructions.

Urine angiotensinogen: The urinary angiotensinogen (uAGT) protein concentration was measured with an ELISA kit (#27412 Human Total Angiotensinogen Assay Kit—IBL, #27413, Fujioka, Japan). Samples were diluted 1:2 for measurements.

### 4.2. Studies in Mice

Development of a high fat diet-induced overweight non-diabetic mouse model: All animal procedures were approved by the Tulane University Animal Care and Use Committee. The 3-week-old male and female C57BL/6 mice 3 were weaned and kept under standard conditions (12-h light/dark cycles and 21 ± 2 °C). All the mice were 4 ± 1–2 weeks of age when the study started. Samples of blood and urine were collected to measure baseline parameters (week 0). At 5 weeks of age, mice were randomly divided into control diet (CD) and high fat diet (HFD), with 4–5 females and males in each group. The CD (PicoLab^®^ Rodent Diet 20 EXT IRR 5R53, St. Louis, MO, USA) consisted of 25% Kcal from protein, 13% from fat, and 62% from carbohydrate. The HFD (TestDiet, DIO 58V8–45, St. Louis, MO, USA) contained 18% Kcal from protein, 45% from fat, and 36% from carbohydrate. Two separate sets of mice were used: one for metabolic purposes (*n* = 5 mice per group) and the other for blood pressure recordings using radiotelemetry (*n* = 4 mice per group). Food and water were ad libitum during the following 8 weeks, until the study was completed.

Urine samples. Urine samples were collected from mice individually housed in metabolic cages and centrifuged at 3500× *g*, 4 °C for 30 min to remove particulate. No endpoint protocol or tissue collection was assigned to this protocol; the endpoint protocol was used for a previous study ending at 38 weeks. Kidney ROS production was measured as previously described for mice tissues in samples of 38 weeks and presented in this manuscript. Superoxide production was quantitatively measured by electron spin resonance spectroscopy, as previously described in detail [17]. The amplitude measurements of the ESR spectra in arbitrary units were normalized to the dry weight of the corresponding tissue samples.

Body weight and glucose measurements: Body weights (BW) from mice fed either a CD or HFD were measured. Blood glucose levels were monitored monthly using a drop of blood obtained by milking the tail and measured with a hand-held glucometer (ONETOUCH Ultra glucometer, LifeScan, catalog #ZJZ8158JT, Milpitas, CA, USA).

Diastolic, Systolic, and Mean Arterial Pressures: Mice were anesthetized by isoflurane inhalation and a left carotid catheter connected to a radiotelemetry transmitter was implanted (TA11-PAC10, Data Sciences Inc., Palo Alto, CA, USA). Blood pressures were recorded in conscious mice after 2 weeks of recovery for one 24 h period each week for four weeks using Ponemah 6.0 software.

Angiotensinogen concentration in urine: The urinary angiotensinogen (uAGT) protein was measured with an ELISA kit (Mouse Total Angiotensinogen Assay Kit, IBL, catalog #27413, Fujioka, Japan). Urinary excretion rates of uAGT were calculated from the 12 h volumes collected.

Albumin: Mouse albumin in urine was measured with ELISA (Abcam, catalog #108792, Billerica, MA, USA).

Statistical analyses: The results are expressed as mean ± SEM. Statistical analyses were performed using GraphPad Prism Software Version 6 (GraphPad Software, Inc., La Jolla, CA, USA). Normal distribution of each parameter analyzed was tested using Shapiro–Wilk. Two-way ANOVA was used to compare the mean differences between groups and divided on variables HFD vs. CD in male and female mice studies or NW or OW in male and female population. Post-test comparisons for two groups by non-paired (one-tailed) *t*-test was also used. The correlation coefficient was used to verify the relationship in the population analyzed. A *p*-value < 0.05 was considered statistically significant. Multiple logistic regression analysis was performed by using predicted versus observed values by categorization, as described in Table 1.

## Figures and Tables

**Figure 1 ijms-25-00635-f001:**
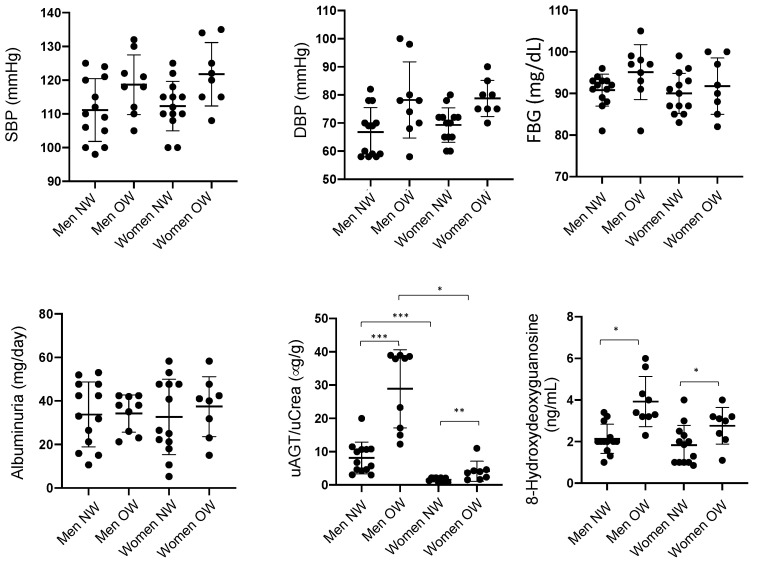
Systolic blood pressure (SBP), diastolic blood pressure (DBP), fasting blood glucose (FBG), albuminuria, urinary angiotensinogen/urinary creatinine (uAGT/uCrea), and urinary levels of reactive oxygen species measured with 8-Hydroxydeoxyguanosine in human subjects with normal weight (NW) or overweight (OW). * *p* < 0.05; ** *p* < 0.01; *** *p* < 0.001. Two-way ANOVA was used to analyze the effects of treatment and sex and multiple comparison analysis was performed to test differences among groups.

**Figure 2 ijms-25-00635-f002:**
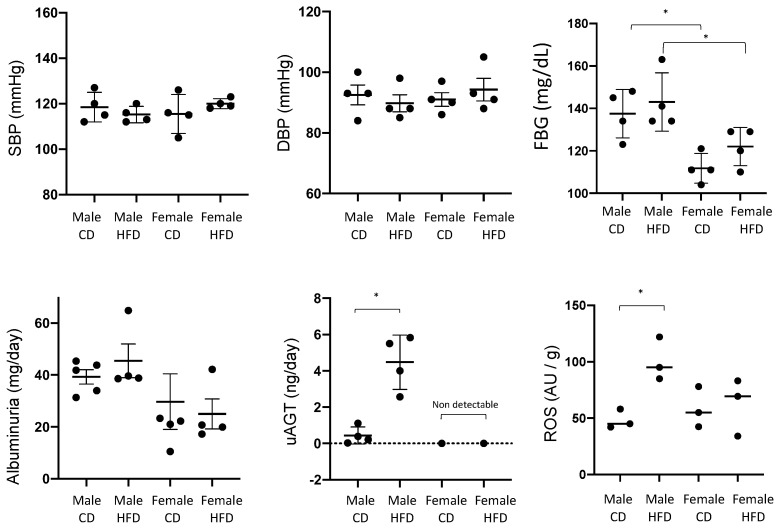
Systolic blood pressure (SBP), diastolic blood pressure (DBP), fasting blood glucose (FBG), albuminuria, and urinary angiotensinogen (uAGT) after 8 weeks of high fat (HFD) or control diet (CD) in male and female mice. Tissue levels of reactive oxygen species (ROS) were measured at 28 weeks (see Section 3 and Section 4). * *p* < 0.05. Two-way ANOVA was used to analyze the effect of treatment and sex and multiple comparison analysis was performed to test differences among groups.

**Table 1 ijms-25-00635-t001:** Inclusion criteria and exclusion criteria.

Inclusion Criteria	Exclusion Criteria
Non-smoking	Smokers
No acute or chronic infection	Acute or chronic infection
No treatment with ARBs, ACE inhibitors (preceding 4 weeks)	Treatment with anti-hypertensive drugs (preceding 4 weeks)
No diabetic treatment (preceding 4 weeks)	Diabetic treatment (preceding 4 weeks)

**Table 2 ijms-25-00635-t002:** Multivariate logistic regression analysis of independent predictors related to uAGT/uCrea levels. CI confidence interval. We performed a multivariate logistic regression analysis to identify independent predictors of uAGT/uCrea ratio. We have included male gender as a predictor, fasting blood glucose (FBG), systolic blood pressure (SBP), diastolic blood pressure (DBP), and overweight.

Variables	Odds Ratio	95% CI	P
Gender (Male?)	1.056	0.8967–1.271	0.4887
FBG ≥ 100 mg/dL	0.893	0.1281–1.948	0.1697
SBP ≥ 120 mmHg	1.008	0.8393–1.194	0.3329
DBP ≥ 80 mmHg	1.056	0.8831–1.2614	0.7410
Albuminuria ≥ 30 mg/g	2.324	1.107–11.83	0.0448 *
Overweight (BMI ≥ 25)	3.933	0.2712–0.7226	<0.0001 *

* Significant association between the predictor and uAGT/uCrea levels.

## Data Availability

The data are available on request.

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
