# Peer review of "Urinary Angiotensinogen Displays Sexual Dimorphism in Non-Diabetic Humans and Mice with Overweight"

_ijms, 2024, doi:10.3390/ijms25010635_

Round 1

Reviewer 1 Report

Comments and Suggestions for Authors

The manuscript examined uAGT excretion in non-diabetic humans and mice with overweight, they concluded that uAGT displays sexual dimorphism in non-diabetic humans and mice with overweight. The reviewer has some comments that need to be addressed.

1. In Studies in humans, the authors only detected urinary AGT mRNA levels but not urinary AGT protein excretion. Where does urinary AGT mRNA originate from, detached renal tubular cells, or others? How about urinary AGT protein levels in these cases? Was there any association between urinary AGT mRNA and urinary AGT protein in these cases? Thus, urinary AGT protein levels should be detected.

2. Studies in humans, the clinical characteristics associated with overweight should be analyzed through univariate and multivariate logistic regression in all subjects.

3. Studies in humans, Urinary levels of kidney injury markers, including NGAL and KIM-1, should be measured, as well as the correlation between Urinary AGT and these markers.

4. Studies in mice, how about urinary AGT mRNA levels in these mice, and the correlation with NGAL, KIM-1, TGF-b, a-SMA, blood pressure, BMI, and albuminuria? These dates should be provided.

Author Response

Reviewer 1 Comments and Suggestions for Authors

The manuscript examined uAGT excretion in non-diabetic humans and mice with overweight, they concluded that uAGT displays sexual dimorphism in non-diabetic humans and mice with overweight. The reviewer has some comments that need to be addressed.

  1. In Studies in humans, the authors only detected urinary AGT mRNA levels but not urinary AGT protein excretion. Where does urinary AGT mRNA originate from, detached renal tubular cells, or others? How about urinary AGT protein levels in these cases? Was there any association between urinary AGT mRNA and urinary AGT protein in these cases? Thus, urinary AGT protein levels should be detected.

Answer: Because this was also suggested by reviewer 3, we have measured uAGT in urine samples from individuals who participated in the study. The results are included in the new version of the manuscript.

  1. In studies in humans, the clinical characteristics associated with being overweight should be analyzed through univariate and multivariate logistic regression in all subjects.

Answer: Many thanks for the suggestion. We have included this analysis in the new version of the manuscript. Please see Table 2.

  1. Studies in humans, Urinary levels of kidney injury markers, including NGAL and KIM-1, should be measured, as well as the correlation between Urinary AGT and these markers.

Answer: We appreciate your suggestion. We did all the analysis mentioned in humans. However, and, as suggested by reviewer 3, when R square is below 0.20, it means the correlation is very weak so it can’t be said correlated. To be consistent and taken in consideration that no significant associations were found, we decided to eliminate this data (mRNA injury markers vs. AGT mRNA) and include the analysis of AGT by ELISA. Similar observations were done by reviewer 3.

  1. Studies in mice, how about urinary AGT mRNA levels in these mice, and the correlation with NGAL, KIM-1, TGF-b, a-SMA, blood pressure, BMI, and albuminuria? These dates should be provided.

Answer: We appreciate your suggestion. Unfortunately, we cannot perform a new set of animals to answer this question at this time since our protocol requires 8 weeks of HFD plus the analysis. Since we have added data of AGT - ELISA in humans by performing a new analysis of human urine samples (showing consistent results), we decided to be concise and consistent by showing similar analysis in mice and humans and do not show the results of the mRNA injury markers in humans since there were not informative and do not show significant correlations among them, this was strongly criticized by reviewer 3.

Reviewer 2 Report

Comments and Suggestions for Authors

The authors look into two different models to understand sexual differences and its effects on angiotensinogen which is interesting to look at. In presence of increased body weight, they are interested in looking at the differences before the onset of diseases at an early stage. There are some edits that need to be made and please find it in the pdf file attached. The authors mention ROS in their introduction as well as discussion but it is not measured in the experiments. It would be interesting to look at the ROS amount in these models. The authors need to add details about the patient inclusion criteria in form of a table. There is not much difference seen in the mice model in the FBG and other characteristic markers, may be there is a need to collect data at a slightly later time point to better address the differences. Specially in females as there in no uAGT detected.  The conclusion statement needs to be edited to better reflect the data. 

Author Response

Reviewer 2 Comments and Suggestions for Authors

The authors look into two different models to understand sexual differences and its effects on angiotensinogen which is interesting to look at. In presence of increased body weight, they are interested in looking at the differences before the onset of diseases at an early stage.

  1. There are some edits that need to be made and please find it in the pdf file attached.

Answer: We really appreciate your help and intention to improve our manuscript.

  1. The authors mention ROS in their introduction as well as discussion but it is not measured in the experiments. It would be interesting to look at the ROS amount in these models.

Answer: Unfortunately, ROS measurements were performed only at 28 weeks in kidney tissues, these results has been added to the new version of the manuscript and discussed accordingly.

  1. The authors need to add details about the patient inclusion criteria in form of a table.

Answer: thank you for your suggestion, we have added this information to the new version of the manuscript. Please see Table 1.

  1. There is not much difference seen in the mice model in the FBG and other characteristic markers, maybe there is a need to collect data at a slightly later time point to better address the differences. Specially in females as there in no uAGT detected. 

Answer: We appreciate your suggestion. Data already published in 2020 by Reverte et al showed that after 28 weeks on the diet, male mice on HFD developed insulin resistance along with the established obesity and a significant elevation in uAGT in males but not females, consistently with our data in non-diabetic animals.

Our protocol uses non-diabetic overweight mice based on our previous study with endpoint at 28 weeks. Increased blood glucose was observed after 16 weeks. To clarify, please see the following data used in previous publication in the pdf attached.

  1. The conclusion statement needs to be edited to better reflect the data.

Answer: Thank you so much, we have edited the manuscript accordingly.

Reviewer 3 Report

Comments and Suggestions for Authors

The authors examined uAGT excretion in non-diabetic humans and mice with overweight (OW) from both sexes and their correlations with BP, body mass index (BMI), and albuminuria and concluded that the urinary excretion of AGT differs between males and females and overcomes overt albuminuria in humans and mice overweight. However, the current results do not support the conclusion. Firstly, the authors mentioned they have examined the human AGT protein level in Abstract, Introduction and Discussion, but we didn’t see it at all in the Result section. Firstly, without the protein level of AGT, there is no meaning for comparing AGT mRNA levels at all. It should be examined by ELISA in human. Secondly, please state clearly the exact changes. For example, in the first paragraph of the Discussion, you can’t state AGT levels are increased in human and mice with OW, but actually it is only increased in MALE, not in female. Thirdly, the author should have examined the mRNA level of AGT in mice since they were analyzed even in human samples. Lastly, for the correlation, when the P value is over 0.05, it means the equation doesn’t exactly fit with the data so R square means nothing with correlations. When the R square is below 0.20, it means the correlation is very weak so it can’t be said correlated.  

There are a few corrections needed in the text as follows:

1.     In the Abstract, the authors didn’t categorize BMI 25. It is currently in both overweight and control groups.

2.     In the Abstract, the authors didn’t say how long to feed the mice with a normal fat diet. It only specifies the HFD.

3.     In the result section, no need to present the exact number of P values and any P value >0.05 should be stated as no changes.

4.     Have the authors tried to see the results in all patients (both male and female)?

5.     In Figure 1, “(E)” is missing.

Comments on the Quality of English Language

It is ok.

Author Response

Reviewer 3 Comments and Suggestions for Authors

The authors examined uAGT excretion in non-diabetic humans and mice with overweight (OW) from both sexes and their correlations with BP, body mass index (BMI), and albuminuria and concluded that the urinary excretion of AGT differs between males and females and overcomes overt albuminuria in humans and mice overweight. However, the current results do not support the conclusion.

Answer: Thank you for your suggestions, we have added a new set of analysis and reworded the conclusions according to the results.

Firstly, the authors mentioned they have examined the human AGT protein level in Abstract, Introduction and Discussion, but we didn’t see it at all in the Result section. Firstly, without the protein level of AGT, there is no meaning for comparing AGT mRNA levels at all. It should be examined by ELISA in human.

Answer: We have done a new single set of experiments by analyzing all urine samples (CAT 27742 Human Total Angiotensinogen Assay Kit – IBL) from individuals who participates in the study. The results are added in results section and replaced the data of mRNA by protein AGT as suggested by referee 1 and 3.

Secondly, please state clearly the exact changes. For example, in the first paragraph of the Discussion, you can’t state AGT levels are increased in human and mice with OW, but actually it is only increased in MALE, not in female.

Answer: We appreciate your suggestions. We have edited the conclusions accordingly.

Thirdly, the author should have examined the mRNA level of AGT in mice since they were analyzed even in human samples.

Answer: We appreciate your suggestion. However, we were unable to perform mRNA analysis at this time. Since we have added data of AGT by ELISA in humans, we decided to be consistent by showing similar analysis in humans and mice (protein by ELISA), also suggested by referee 1.

Lastly, for the correlation, when the P value is over 0.05, it means the equation doesn’t exactly fit with the data so R square means nothing with correlations. When the R square is below 0.20, it means the correlation is very weak so it can’t be said correlated.  

Answer: We really appreciate your suggestion to improve our manuscript. Since reviewer 1 also suggested to perform an analysis of uAGT by ELISA, end because no significant associations were found when analyzing mRNA, we decided to replace this data by a new analysis in urine that includes uAGT by ELISA and multivariate logistic regression in all subjects.

There are a few corrections needed in the text as follows:

  1. In the Abstract, the authors didn’t categorize BMI 25. It is currently in both overweight and control groups.

Answer: Thank you for your comment, this information has been corrected in the new version of the manuscript.

  1. In the Abstract, the authors didn’t say how long to feed the mice with a normal fat diet. It only specifies the HFD.

Answer: thank you for your comment, this information was added.

  1. In the result section, no need to present the exact number of P values and any P value >0.05 should be stated as no changes.

Answer: Thank you for your comments, we have edited the manuscript accordingly.

  1. Have the authors tried to see the results in all patients (both male and female)?

Answer: Many thanks for your input. We perform a multivariate logistic regression analysis to identify independent predictors of uAGT/UCrea values in the human subjects. We included gender (male or not) as a predictor, fasting blood glucose (FBG) less than 100 mg/dL; systolic blood pressure (SBP) less than 120 mm Hg, diastolic blood pressure (DBP) less than 80 mm Hg and overweight (BMI 25), please see Table 1.

  1. In Figure 1, “(E)” is missing.

Answer: Thank you for your information, we have replaced all figures according to the comments of reviewers 1 and 3.

Round 2

Reviewer 1 Report

Comments and Suggestions for Authors

I have no further comments.

Author Response

Thank you for your comments and sugestions.